# Factors Influencing the Properties of Extrusion-Based 3D-Printed Alkali-Activated Fly Ash-Slag Mortar

**DOI:** 10.3390/ma15051969

**Published:** 2022-03-07

**Authors:** Qiang Yuan, Chao Gao, Tingjie Huang, Shenghao Zuo, Hao Yao, Kai Zhang, Yanling Huang, Jing Liu

**Affiliations:** 1School of Civil Engineering, Central South University, Changsha 410075, China; yuanqiang@csu.edu.cn (Q.Y.); gc1977716234@163.com (C.G.); zuosh93@csu.edu.cn (S.Z.); yaohao@csu.edu.cn (H.Y.); zkai2766@163.com (K.Z.); 2National Engineering Research Center of High-Speed Railway Construction Technology, Changsha 410075, China; 3Fujian Strait Environmental Protection Group Co., Ltd., Fuzhou 350014, China; huangyl15111337148@163.com; 4China Academy of Railway Sciences Railway Engineering Research Institute, Beijing 100081, China

**Keywords:** extrusion-based 3D printing, geopolymer, printability, interlayer bond strength, drying shrinkage, mix proportioning

## Abstract

The mix proportioning of extrusion-based 3D-printed cementitious material should balance printability and hardened properties. This paper investigated the effects of three key mix proportion parameters of 3D-printed alkali-activated fly ash/slag (3D-AAFS) mortar, i.e., the sand to binder (s/b) ratio, fly ash/ground granulated blast-furnace slag (FA/GGBS) ratio, and silicate modulus (Ms) of the activator, on extrudability, buildability, interlayer strength, and drying shrinkage. The results showed that the loss of extrudability and the development of buildability were accelerated by increasing the s/b ratio, decreasing the FA/GGBS ratio, or using a lower Ms activator. A rise in the s/b ratio improved the interlayer strength and reduces the drying shrinkage. Although increasing the FA/GGBS mass ratio from 1 to 3 led to a reduction of 35% in the interlayer bond strength, it decreased the shrinkage strain by half. A larger silicate modulus was beneficial to the interlayer bond strength, but it made shrinkage more serious. Moreover, a simple centroid design method was developed for optimizing the mix proportion of 3D-AAFS mortar to simultaneously meet the requirements of printability and hardened properties.

## 1. Introduction

3D printing, also known as additive manufacturing, is an emerging technology to produce well-designed 3D products in the absence of formwork [1,2]. It has made great progress in industrial production, aerospace technology, biomaterials, medicine and health, the automobile industry, and the electronic industry [3,4,5,6,7]. In recent years, 3D printing has also been applied to the field of civil engineering [8,9,10,11,12], mainly including powder-based 3D printing (D-Shape) and extrusion-based 3D printing (Contour Crafting and Concrete Printing) [13,14,15,16,17,18]. With the help of computer-aided design and automatic operation, 3D printing has the characteristics of flexible design, fast construction speed, and low labor and energy consumption [19,20,21,22,23,24]. It is conceivable that this technology has great potential for large-scale application in civil engineering.

Unlike the conventional building process, 3D-printed concrete is a from-work free construction technique. Hence, the first challenge of this technology is to prepare concrete materials that are compatible with printing technology. That is, the mixture needs good fluidity and self-supporting capacity to ensure that the continuous concrete can be extruded through the nozzle and bears the load generated by subsequent concrete layers [25,26]. The printability of concrete is mainly characterized by extrudability and buildability [17]. Currently, due to the lack of relevant standards for 3D printing concrete, researchers often use standardized methods in the field of cement-based materials or even other fields to characterize the printability of 3D printing cement-based materials. For example, the drop test, slump test, T50 slump test, V-funnel test, and L-box test [27,28,29,30,31] are usually used to evaluate the flow behavior of 3D-printed concrete. The deformation of printed objects, such as longitudinal deformation or cross-sectional area deformation, is used to characterize buildability [27,30,32,33,34]. In addition, wet embryo strength and cylinder stability tests are other methods for evaluating the buildability of materials under loading [27,35]. Generally, a good 3D-printed concrete should be a thixotropic fluid. Under the action of shear, the static yield stress and plastic viscosity of these materials are reduced for easier extrusion. Once printed, the material at rest should recover its static yield stress to maintain its shape [36,37,38].

Portland cement is the most widely used binder for 3D-printed concrete. However, due to the high energy consumption and high carbon dioxide emissions in its production process [39,40,41,42], Portland cement no longer meets the needs of the times. In response to the global call for sustainable development, finding alternative binders for Portland cement with low environmental impact has been the top task in the field of cement and concrete. Among these, geopolymers, or alkali-activated materials, are considered to be the most likely alternative in the 21st century due to their low energy consumption, low carbon dioxide emission, high compressive strength and good durability [39,42,43,44]. This kind of sustainable material can be utilized in the construction of areas rich in waste/byproducts, and some studies have been focusing on 3D-printed geopolymers [10,11,30,45,46,47,48].

Geopolymers consist of various precursors and activators, which are more variable in composition than Portland cement. Thus, its properties are more flexible and can be tailored for different purposes. Alghamdi et al. [45] proposed that the chemical composition, type, and concentration of the activator are important parameters affecting the printability of 3D-printed geopolymer pastes. The literature [30] showed that an alkaline activator with a high Si/Na ratio reduced the viscosity and yield stress growth rate of fresh paste, thus affecting the extrudability and buildability of the material. Studies [10,11,34,49] have shown that the addition of slag, lime, clay, fiber, and silica fume can improve the buildability of 3D-printed fly ash-based geopolymers. Chougan et al. [50] found that adding 0.3% and 1% nano graphite platelets (NGPs) reduced the workability of the sample due to thickening; however, the addition of 0.1% and 0.5% NGPs showed a lubrication effect and improved the workability of the geopolymer. Furthermore, the relationship between the printability and rheological parameters of 3D-printed geopolymers was also studied. Le et al. [17] proposed that yield stresses of 178.5–359.8 Pa are required for 3D printing mortar, while Panda and Tan [34] found that a yield stress of 600–1000 Pa was the favorable scope of mortar extrusion.

Since 3D printing is a process printing material layer by layer, the interlayer bond strength between layers is the weak part of the printed element. Interlayer bonding is a major concern of 3D printing technology. The interlayer bond strength may be measured by compressive strength and flexural strength, tensile strength, splitting tensile and slant shear tests [51,52,53,54]. It was found that adding fibers to geopolymers can reduce interlaminar deformation, but an increase in porosity will lead to a decrease in the interlayer bond strength [55,56]. The use of slag contributed to the higher compressive strength of 3D-printed fly ash-based geopolymers [11,49], and steel cables can increase the flexural strength of 3D-printed geopolymer composites by 290% [57]. Additionally, with increasing printing speed, the tensile strength decreases slightly [58].

Although geopolymers have the advantages of high strength, high-temperature resistance, acid corrosion resistance, and good permeability resistance, they have the problem of volume stability [11,59,60]. Collins et al. [59] found that the shrinkage of slag-based geopolymers is more than three times larger than that of Portland cement after 60 d of curing. However, there are limited studies on the volume stability of 3D printing hardened geopolymers.

Trial and error or single factor variable methods are often used to optimize the mix proportion of 3D-printed concrete. Obviously, these design methods cannot take printability, interlayer bonding, and volume stability into consideration simultaneously. These three properties are the most important properties for 3D-printed geopolymers. In addition, the properties of geopolymers may vary greatly due to the regional nature of geopolymer raw materials. This brings much more difficulties to the design of 3D-printed geopolymers. The interlayer bond strength and volume stability, which are also extremely important for the engineering application of 3D printing materials, are usually not considered in mix proportioning. Therefore, it is necessary to develop a mix proportioning method of 3D printing geopolymer mixtures, which can take printability, interlayer bond strength, and volume stability into consideration at the same time. This will greatly help to promote the application of this eco-friendly material in 3D printing technology.

For a full understanding of the relationship between the composition of 3D-printed alkali-activated fly ash/slag (3D-AAFS) mortar and its properties, this study comprehensively investigated the influences of the sand-to-binder ratio, the relative proportion of FA-GGBS precursors, and the silicate modulus of the activator on the printability, interlayer bond strength, and volume stability of 3D-AAFS mortar for the first time. Moreover, a simple centroid design method was developed for mix proportioning of extrusion-based 3D-AAFS mortar to strike a balance among printability, interlayer bond strength, and volume stability. This study enriched the mix design concept for 3D-printed alkali-activated materials.

## 2. Experimental Program

### 2.1. Raw Materials and Sample Preparation

In this study, Grade I fly ash (FA) powder in compliance with GB/T 1596-2017 [61] and Grade 95 ground granulated blast furnace slag powder (GGBS) in accordance with GB/T 18046-2017 [62] were used to prepare 3D-printed alkali-activated fly ash/slag (3D-AAFS) mortars. Their chemical compositions were determined by X-ray fluorescence (Axios mAX, PANalytical, Almelo, The Netherlands) and are given in Table 1. The density and Blaine specific surface area were 2350 kg/m^3^ and 263 m^2^/kg for fly ash and 2860 kg/m^3^ and 487 m^2^/kg for GGBS. The fine aggregate used was river sand with a fineness modulus of 2.47. The particle size distributions of powder materials and sand are shown in Figure 1. The alkaline activator was prepared using sodium hydroxide pellets (analytical grade, purity ≥ 98%), liquid sodium silicate with an original silicate modulus (Ms = SiO_2_/Na_2_O) of 3.1 (water content of 62%), and distilled water to achieve different silicate moduli (i.e., 0, 0.5, and 1).

3D-AAFS mortars with a water-to-binder (w/b) ratio of 0.35 and sand-to-binder (s/b) ratio varying from 0.8 to 1.2 were prepared. The water contained in liquid sodium silicate was considered in the total mixing water. The mix proportions of 3D-AAFS mortars are presented in Table 2. According to the designed silicate modulus shown in Table 2, the alkaline solution was prepared 2 h before the experiment and cooled to room temperature (25 ± 3 °C). The powder materials were mixed thoroughly in a Turbula shaker (WAB AG, Basel, Switzerland) for 24 h. Before preparing the 3D-AAFS mortars, the dry-mixed powder and sand were introduced into the mixer, and then the alkaline solution was added and stirred at 500 rpm for 4 min.

### 2.2. Rheological Tests for AAFS Mortar

Rheological tests were performed by a Rheolab QC rheometer (Anton Paar, Graz, Austria) with a cylindrical geometry of 41.94 mm in inner diameter. The type of rotator was ST22-4 V-40-SN20452, and the height and width of each blade were 40.00 mm and 22.00 mm, respectively. During the test, the temperature was kept at 25 °C using a water bath.

#### 2.2.1. Dynamic Test

The procedure of the dynamic yield stress test consisted of preshearing at 100 s^−1^ for 60 s, resting for 15 s, ramping up from 0 to 100 s^−1^ in 60 s, stopping for 30 s, and then ramping down from 100 to 0 s^−1^ in 60 s. The Herschel–Bulkley model (H–B model, Equation (1)) was used to characterize the rheological behavior of the samples.
(1)$$\tau =\tau_{d,\;0}+K{\dot{\gamma }}^{n}$$
where τ is the shear stress (Pa), $\tau_{d,0}$ is the dynamic yield stress (Pa), *K* is the consistency coefficient (Pa·s), and n is the dimensionless fluidity index.

#### 2.2.2. Static Test

Before the static yield shear test, the sample was presheared at 100 s^−1^ for 60 s to reach a consistent initial state [63]. The test was run at a constant shear rate of 0.02 s^−1^ for 60 s, the peak value of the measured shear stress was denoted as the static yield stress. The test was repeated every 5 min until the yield stress reached the upper limit value of the instrument. In this paper, the model (Equation (2)) proposed by Roussel [64] was used to fit the static yield stress data, which was defined as the measured peak shear stress.
(2)$$\tau_{0}\left(t\right)=\tau_{0,0}+A_{\mathrm{thix}}t$$
where $\tau_{0,0}$ and *A*_thix_ are the static yield stress and the structural build-up rate, respectively.

### 2.3. Printability Tests

#### 2.3.1. Flow Table Test

The flowability of the AAFS mortar was evaluated by using a flow table test on the basis of ASTM C230 [65]. All specimens were tested every 10 min after mixing. It has been proven that the spread diameter determined by the flow table test is related to its pumpability and extrudability [34,66,67]. In this work, it was found that the mixture with a spread diameter less than 200 mm cannot be continuously extruded from the 3D printer.

#### 2.3.2. Buildability Test

A self-designed device was used to measure the buildability of 3D-AAFS mortar. The details of the device and operations information can be found in our previous work [33]. In this method, the demolded AAFS mortar specimen with a diameter of 50 mm was subjected to a load, which equaled 20 times the weight of the specimen. The deformation under this 20-layer load was recorded and used to evaluate the buildability. The buildability of mortar is acceptable when its deformation is smaller than 0.2% [33].

### 2.4. Preparation of 3D-AAFS Mortar Specimen

For mortar printing, a lab-scale 3D concrete printer with a round nozzle of 30 mm diameter, introduced by earlier works [31,63,64], was employed in this study. In this work, the printing speed was set at 30 mm/s, and the printing interval time was controlled at 5 min. All the samples for mechanical tests and drying shrinkage tests were cut from the printing specimen. The 300 × 75 × 75 mm^3^ blocks were printed. Immediately after printing, both ends of the printed block were cut off, and the middle part of block (285 × 75 × 75 mm^3^) was used for the drying shrinkage test, and the sample was cured at 25 ± 2 °C and 50 ± 5% relative humidity. For the interlayer bond strength test, the printed blocks were cured at 25 °C and 98% relative humidity for 2 days and then cut into 70 mm × 70 mm × 70 mm cubes. The cubes were cured at 25 °C and 98% relative humidity.

### 2.5. Interlayer Bond Strength Test

The interlayer bond strengths of the specimen at 3, 7, 28, and 90 days were measured by the slant shear bond strength, as shown in Figure 2. In the slant shear strength test, the load was directly applied to the mold, and the inclined angle of the specimen was set to 60 degrees [68,69]. The loading rate was 2.4 kN/s. The interlayer shear stress (*τ*) was calculated by Equation (3).
(3)$$\tau =\frac{Psin60{}^{\circ}}{A}$$
where *τ* (Pa) is the interfacial shear stress, *P* (N) is the critical load of interface sliding, and *A* is the bond area of the specimen.

### 2.6. Drying Shrinkage

The drying shrinkage test in this study was carried out according to ASTM C157 [70]. For the measurement of drying shrinkage strains, the 3D AAFS mortar specimen was placed in the shrinkage frame in the vertical position, and an electronic dial gauge was mounted on the top, as shown in Figure 3. The specimens were kept at a controlled temperature of 25 ± 2 °C and relative humidity of 50 ± 5%. The dial gauge reading was recorded at the age of 1, 2, 3, 5, 7, 9, 11, 14, 21, 28, 56, and 90 days. The shrinkage strain of the mix reported in the paper is the average of three specimens.

## 3. Results and Discussion

### 3.1. Rheological Behavior

Figure 4a–c show the effects of the s/b ratio, FA/GGBS mass ratio, and silicate modulus of alkaline solution on the dynamic yield stress (*τ*_d,0_) of AAFS mortar. The *τ*_d,0_ of mortar corresponds to the stress required for initiating flow [71]. Figure 4a reveals that the *τ*_d,0_ value of AAFS mortar is sensitive to the s/b ratio. The rise of the s/b ratio leads to a significantly higher *τ*_d,0_. As seen, the *τ*_d,0_ of P1-S1.2-M0 is nearly twice the value of P1-S0.8-M0. The increase of s/b ratio results in a higher solid volume fraction and the increase of particle friction, which is mainly responsible for the greater dynamic yield stress [71]. Moreover, Figure 4b indicates that the FA/GGBS ratio also affects *τ*_d,0_ of the mortar. The value shows a slight decline when the FA/GGBS ratio increases from 1 to 3 due to the spherical geometry and smooth surface of FA particles [71,72,73,74]. Figure 4c shows that *τ*_d,0_ has a 40% reduction when the silicate modulus of the activator solution increases from 0 to 0.5. This suggests that the presence of sodium silicate in the activator solution greatly lowers *τ*_d,0_ compared to the AAFS mortar activated only by NaOH (P1-S1-M0). An increase in the silicate modulus in the NaOH + sodium silicate activator solution can further reduce *τ*_d,0_ of the AAFS mortar. It is attributed to the stronger repulsive electrostatic force of particles caused by the increase of silicate modulus of activator solution [72,73].

Figure 5 presents the time evolution of the static yield stress of the AAFS mortar. During the measurement period, the static yield stress values of all samples increase linearly with time. Therefore, Roussel’s model (Equation (2)) was employed to fit these data, and the results are given in Table 3. The results show that mortar with a larger s/b ratio has a greater *A*_thix_, indicating that a higher s/b ratio is beneficial for the structural buildup of AAFS mortar. By comparing the *A*_thix_ values of P1-S0.8-M0, P2-S0.8-M0, and P3-S0.8-M0, it can be found that the increase in FA proportion dramatically slows down the structural buildup rate of mortar. The *A*_thix_ value of the mortar with an FA/GGBS mass ratio of 3 is only one quarter of the value for the mortar with a ratio of 1 due to the lower dissolution rate of the glassy structure of FA as compared to GGBS [74]. In addition, the structural buildup rates of those activated by NaOH+sodium silicate are much slower than those activated by NaOH. Increasing the silicate modulus of the activator from 0.5 to 1 slightly decreased *A*_thix_. This means that the increase in silicate modulus is unfavorable for the structure formation of AAFS mortar, which is owing to the retardation effect of the larger Ms on the formation of reaction products [73,74].

### 3.2. Printability

Figure 6a,b exhibit the changes in fluidity and buildability of AAFS mortar with time, respectively. The spread diameter measured by the flow table has been proven to be useful for predicting the extrudability of cementitious material [66,67]. A larger spread diameter corresponds to a better extrudability [66]. It can be seen from Figure 6a that the initial spread diameters of all mortars (at 5 min) are larger than 200 mm, and all of them can be continuously extruded from the nozzle. The initial spread diameter of the AAFS mortar increases with decreasing s/b ratio, increasing FA/GGBS ratio, and increasing silicate modulus of the alkaline solution. As seen in Figure 7, the spread diameter of mortar shows a good negative correlation with its dynamic yield stress. Therefore, the effects of these three factors on the dynamic yield stress of AAFS mortar should be responsible for their influences on the initial extrudability. Moreover, the spread diameter of mortar with a larger s/b ratio, a lower FA proportion, or a smaller silicate modulus of activator solution reduces faster to be smaller than 200 mm (the critical value corresponding to the acceptable extrudability) earlier.

Figure 6b shows that the initial deformations of all AAFS mortars are greater than 0.2%, which indicates that the buildability levels of all specimens are not acceptable. The deformation value decreases with time, reflecting the improvement of buildability. The increase in the s/b ratio has a positive effect on the buildability of mortar, and mortar with a high sand content reaches an acceptable buildability quicker. The buildability shows an insignificant difference as the FA/GGBS mass ratio increases from 1 to 2 during the whole measurement period. However, the buildability of mortar with an FA/GGBS ratio of 3 is slightly poorer than that for a lower FA/GGBS ratio. The increase in the silicate modulus of the activator solution greatly harms the initial value and the development rate of buildability. Figure 8 plots the static yield strengths of AAFS mortars vs. their deformation. The result shows that the buildability of mortar correlates well to its static yield strength. A higher static yield strength responds to a greater capacity to resist the weight of subsequently extruded layers. The effects of the s/b ratio, FA/GGBS ratio, and silicate modulus on the buildability should be attributed to their effects on the static yield stress. According to the result shown in Figure 8, the buildability of AAFS mortar is acceptable when the static yield stress is larger than ~3.7 kPa. The higher structuration rate (*A*_thix_) resulted from an increase in the s/b ratio, a decrease in the FA/GGBS ratio, or a decrease in the silicate modulus, which accelerated the development of AAFS mortar buildability to meet the requirements of 3D printing manufacturing.

In this study, t_E_ was defined as the duration time AAFS mortar maintains extrudability, which can be estimated by the time the spread diameter becomes smaller than 200 mm. The t_B_ was defined as the needed time for obtaining good buildability, corresponding to the time at which the deformation under a 20-layer load decreased to less than 0.2%. Table 4 lists the t_E_ and t_B_ of AAFS mortars estimated from Figure 6a,b. Both t_E_ and t_B_ determine the open time for the 3D printing process. The total operation time of 3D printing should be shorter than the t_E_ owing to the extrudability requirement. Moreover, to ensure that the bottom layers, especially the first layer, can endure the weight of subsequent layers being deposited on the top without excessive distortion and failures, the start time of 3D printing is suggested to be later than t_B_. Table 4 shows that an increase in the s/b ratio not only shortens t_E_ but also reduces t_B._ Increasing the FA/GGBS ratio of the binder or the silicate modulus of the activator extends the t_E_ and prolongs the t_B_. Increasing the FA/GGBS ratio of the binder or the silicate modulus of the activator retards the reaction products formation, which reduces the structural buildup of mortar and thus extends the t_E_ and prolongs the t_B_, as seen in Table 4.

### 3.3. Interlayer Bond Strength

The developments of the interlayer bond strength of the 3D-AAFS mortars are shown in Figure 9. A rapid increase in the interlayer bond strength of 3D-AAFS mortar can be seen at early stages. Most of the mortars reach more than 60% of the strength of 90 days within 7 days. The early interlayer bond strength is enhanced with an increase in the s/b ratio, and an increase in the s/b ratio from 0.8 to 1.2 leads to an increase in the strength of 1 MPa at 7 days. However, the influence of the s/b ratio on the interlayer strength of 3D-AAFS mortar at 90 days is very small. Conversely, the FA/GGBS mass ratio shows a significant impact on the interlayer bond strength of the printed specimen. The strength of the specimen has a 50% reduction at 7 days and a 40% reduction at 90 days due to the increase in the FA/GGBS ratio from 1 to 3. Figure 9c reveals that the printed mortar with a greater silicate modulus has a much higher interlayer bond strength after 7 days. As the silicate modulus increases from 0 to 1, the strength of the printed specimen is finally increased by ~2 MPa.

In summary, the increase in the s/b ratio only improves the interlayer bond strength at the early stage, but it insignificantly influences the final strength. Increasing the FA/GGBS mass ratio negatively affects the interlayer bond strength, while enlarging the silicate modulus is positive to improve the strength.

### 3.4. Drying Shrinkage

Figure 10a–c display the changes in the drying shrinkage strain of 3D-AAFS mortars with time. The results demonstrate that the drying shrinkage of the printed specimen greatly depends on the s/b ratio and FA/GGBS mass ratio. The effect of the silicate modulus on the drying shrinkage is relatively slighter than that of the other factors. The strains caused by drying shrinkage for P1-S0.8-M0 and P1-S1-M0 were similar within 90 days. This means that increasing the s/b ratio from 0.8 to 1 does not influence the drying shrinkage degree of mortar. However, further increasing the s/b ratio to 1.2 effectively reduces the drying shrinkage after 14 days. At 90 d, the drying shrinkage strain decreased from 0.12% to 0.08%. The 3D-AAFS mortar with a larger FA content shows a clear advantage in reducing drying shrinkage. Figure 10b shows that the dry shrinkage of the printed specimen with an FA/GGBS ratio of 1 is nearly 1.5 times larger than that for an FA/GGBS ratio of 2 and three times greater than that for an FA/GGBS ratio of 3 after 21 days. The variation in silicate modulus barely changes the drying shrinkage level of 3D-AAFS mortar before 28 days. After 28 days, the drying shrinkage increases slightly with increasing silicate modulus.

### 3.5. Mixture Design of 3D-AAFS Mortar Using the Simplex Centroid Design Method

Generally, the sand-to-binder ratio, the mass ratio of FA/GGBS, and the silicate modulus of the activator are the three most important parameters considered in the mix design process of alkali-activated fly ash/slag materials. These three parameters have been proven by many studies [75,76,77,78] to determine the workability, mechanical properties, and durability of mixtures for traditional engineering applications.

The mix design of alkali-activated fly ash/slag materials for 3D printing manufacturing is more complex than that for conventional applications. Printable alkali-activated fly ash/slag materials should meet extra fresh-state and hardened-state requirements, including pumpability, extrudability, buildability, and interlayer strength. Moreover, the drying shrinkage of alkali-activated materials is more serious than that of Portland cement-based materials [78]. The cracks caused by drying shrinkage will threaten the appearance, quality, and even safety of 3D-printed alkali-activated material construction. Therefore, drying shrinkage should be a crucial hardened-state requirement if alkali-activated materials are used for 3D printing.

According to Section 3.2, Section 3.3 and Section 3.4, the mix proportion parameters, including the s/b ratio, FA/GGBS ratio, and silicate modulus, exert great influences on the printability, interlayer bond strength, and drying shrinkage of the 3D-printed alkali-activated fly ash/slag mortar. An increase in the s/b ratio shortens t_E_ and t_B_, improves the interlayer bond strength, and reduces the drying shrinkage of the 3D-AAFS mortar. The rise of the FA/GGBS mass ratio of the binder leads to longer t_E_ and t_B_, a weaker interlayer bond strength, and a smaller drying shrinkage. In addition, the increase in silicate modulus of the activator extends the t_E_, prolongs the t_B_, enhances the interlayer bond strength, and enlarges the drying shrinkage.

Given that all the properties of 3D-AAFS mortar are closely related to the three mix proportion parameters, it is difficult to optimize the mix proportion to satisfy all the requirements. The simple-centroid design method [79] allows us to investigate the properties of mixtures simultaneously controlled by three factors.

In the simple-centroid design method, if there are *n* variables, 2*n*−1 groups of tests are needed to obtain the corresponding contour map. For 3D-AAFS mortar, the factors s/b ratio, FA/GGBS ratio, and silicate modulus are independent of each other. Therefore, a linear substitution calculation method is proposed, which transforms the actual value of variables into the equivalent value with the sum of 100%. The corresponding calculation formulas and values are shown in Table 5 and Equations (4)–(6). Figure 11 describes the seven test points of the simplex centroid design method with three variables, namely, the vertex, midpoint, and center of the triangle.
(4)$$x_{1}+x_{2}+x_{3}=1$$
(5)$$\left[\begin{array}{c} x_{1}\\ x_{2}\\ x_{3} \end{array}\right]=X\left[\begin{array}{c} z_{1}\\ z_{2}\\ z_{3} \end{array}\right]$$
(6)$$X=\left[\begin{array}{ccc} \frac{z_{1}-z_{1min}}{\Delta z_{1}} & 0 & 0\\ 0 & \frac{z_{2}-z_{2min}}{\Delta z_{2}} & 0\\ 0 & 0 & \frac{z_{3}-z_{3min}}{\Delta z_{3}} \end{array}\right]$$
where $x_{i}$, $z_{i}$ and $\Delta z_{i}$ are the equivalent value, the actual value of the variable, and the maximum difference between actual values, respectively.

Based on the experimental results of the properties shown in Table 6, the contours of t_E_, t_B_, and 90-day interlayer bond strength and 90-day drying shrinkage of 3D-AAFS mortar related to the s/b ratio, FA/GGBS mass ratio, and silicate modulus are drawn and shown in Figure 12a–d. These contours can not only be used to predict the performances of 3D-AAFS mortar through its mix proportion, but can also be used to optimize the mix proportion to achieve the designed properties. For example, the interlayer bond strength of 3D printing material was suggested to be higher than ~6 MPa, and the dry shrinkage of alkaline-activated materials should be lower than ~0.09% [43,54,56,60,80,81,82,83]. In practice, t_E_ is always required to be larger than 1 h [29,80]. From the contours of each property, three critical lines of each property could be acquired to meet these required values, as shown in Figure 13. The overlapping area of three different areas in Figure 13 is regarded as the optimal mix proportion of AAFS mortar for 3D printing manufacturing, i.e., the optimum s/b ratio, FA/GGBS mass ratio, and silicate modulus are 0.8~1.0, 2.2~2.6, and 0.5~1.0, respectively.

## 4. Conclusions

The following conclusions can be summarized from the presented findings:(1)The composition of alkali-activated fly ash/slag (AAFS) mortar exerts a tremendous influence on the printability of AAFS mortar, which relates closely to the rheological properties. The printability of AAFS mortar relates closely to the rheological properties. The increase in the s/b ratio enlarges the dynamic yield stress and accelerates the structural buildup of mortar, resulting in a faster loss rate of extrudability and a quicker growth rate of buildability. Conversely, increasing the FA/GGBS mass ratio or the silicate modulus reduces both the dynamic yield stress and structuration rate, which extends the duration time of extrudability and slows down the development of buildability. The printability of AAFS mortar is most sensitive to the silicate modulus of the activator.(2)The hardened-state properties of 3D-printed AAFS mortar also depend on its mix proportion. Increasing the s/b ratio is conducive to improving the interlayer bond strength and diminishing the drying shrinkage. The rise in the FA/GGBS mass ratio weakens the interlayer bond strength and reduces the drying shrinkage. The use of an activator with a larger silicate modulus is beneficial to the interlayer bond strength, but it causes a slightly larger drying shrinkage.(3)A simple centroid design method was developed for mix proportioning of extrusion-based 3D-printed AAFS mortar for the first time, which took printability, interlayer bond strength, and drying shrinkage into consideration at the same time. By restricting the fresh-state and hardened-state requirements, the optimum mix proportion of 3D-AAFS mortar can be obtained using this method.

## Figures and Tables

**Figure 1 materials-15-01969-f001:**
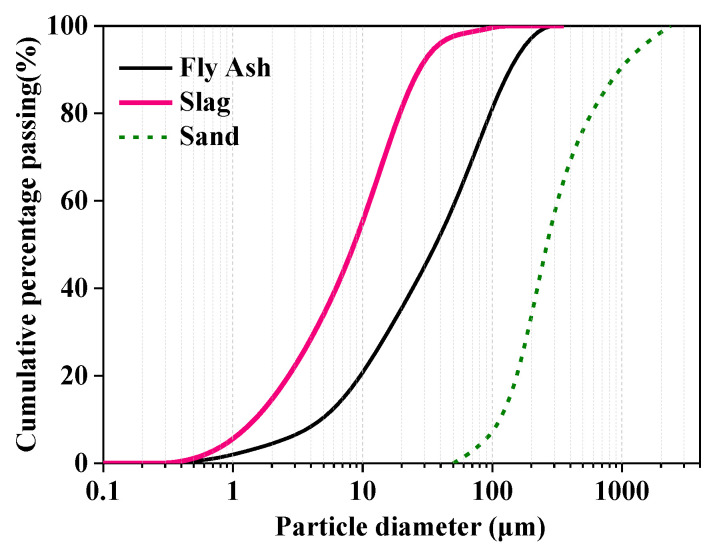
Particle size distributions of powder materials.

**Figure 2 materials-15-01969-f002:**
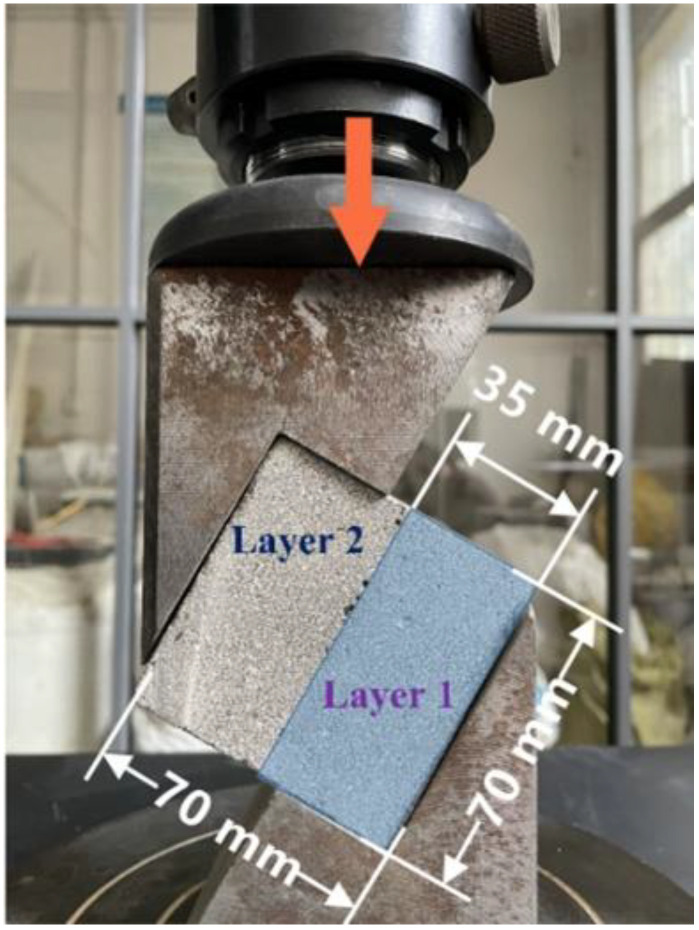
Measurement setup for slant shear test.

**Figure 3 materials-15-01969-f003:**
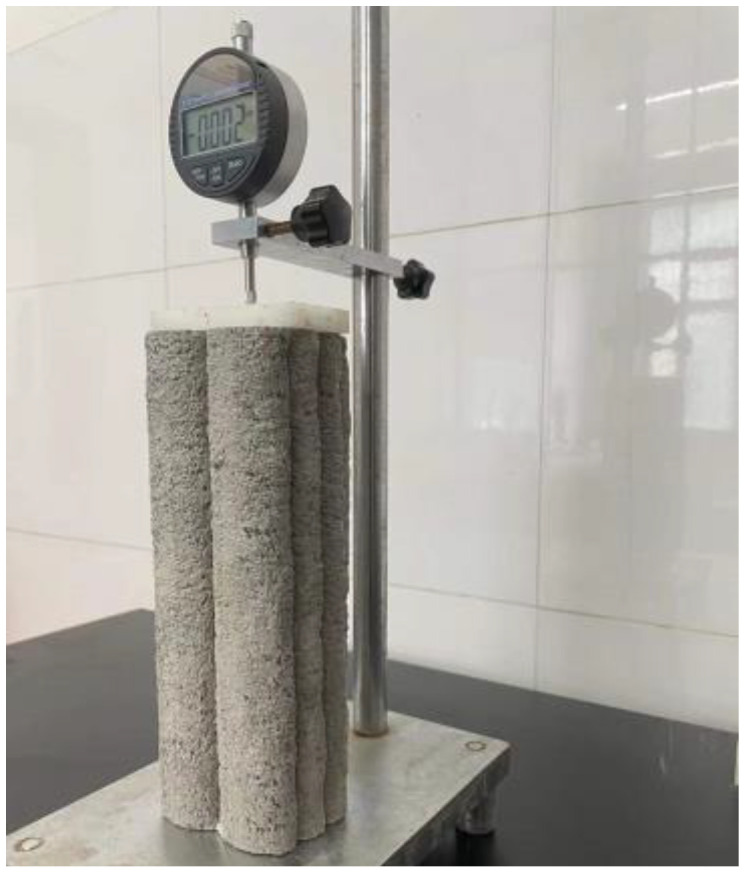
Measurement setup for the drying shrinkage test.

**Figure 4 materials-15-01969-f004:**
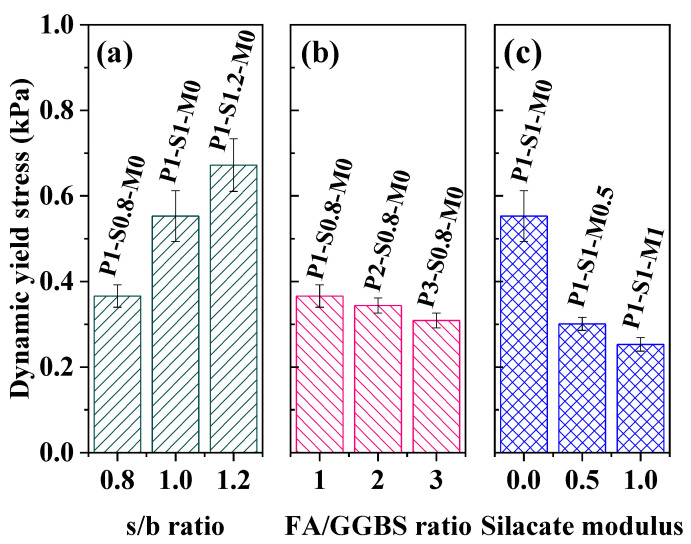
The dynamic yield stress of AAFS mortar after mixing. (**a**) effect of s/b ratio, (**b**) effect of FA/GGBS mass ratio, (**c**) effect of silicate modulus.

**Figure 5 materials-15-01969-f005:**
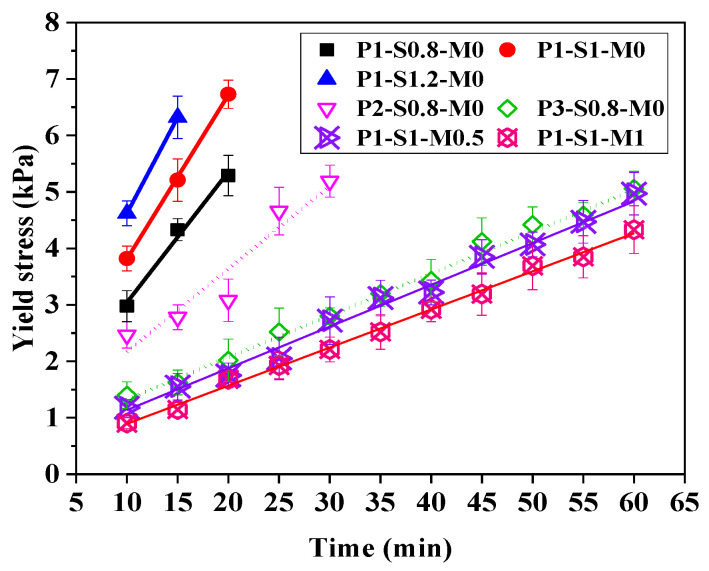
Evolution of static yield stress of samples with time.

**Figure 6 materials-15-01969-f006:**
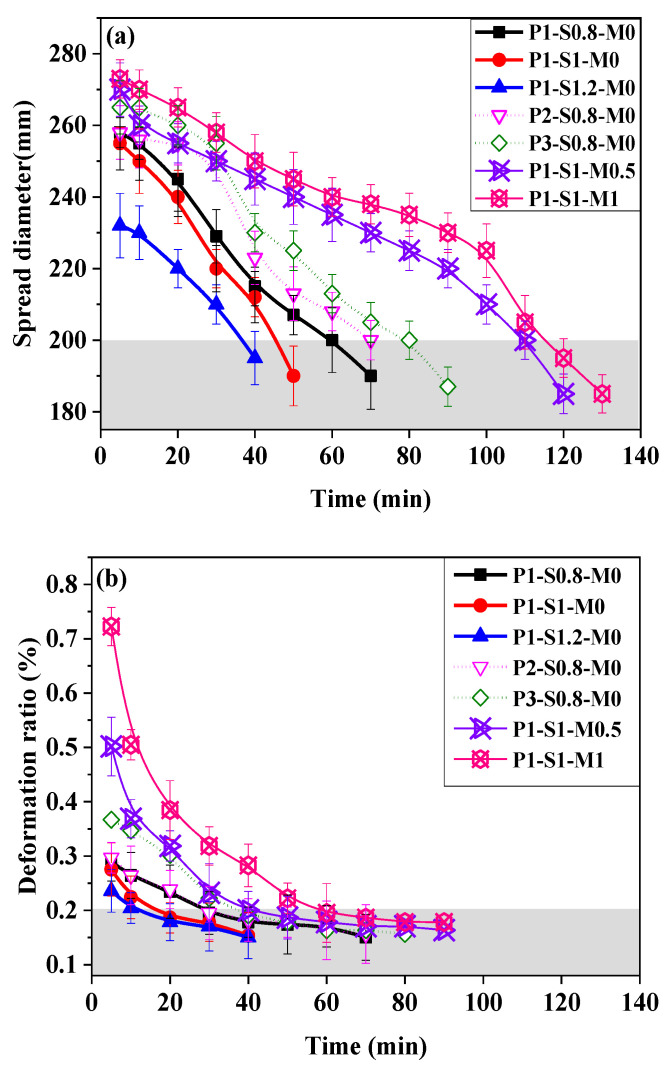
The change of printability of AAFS mortar with time. (**a**) Flow table test result and (**b**) buildability test result.

**Figure 7 materials-15-01969-f007:**
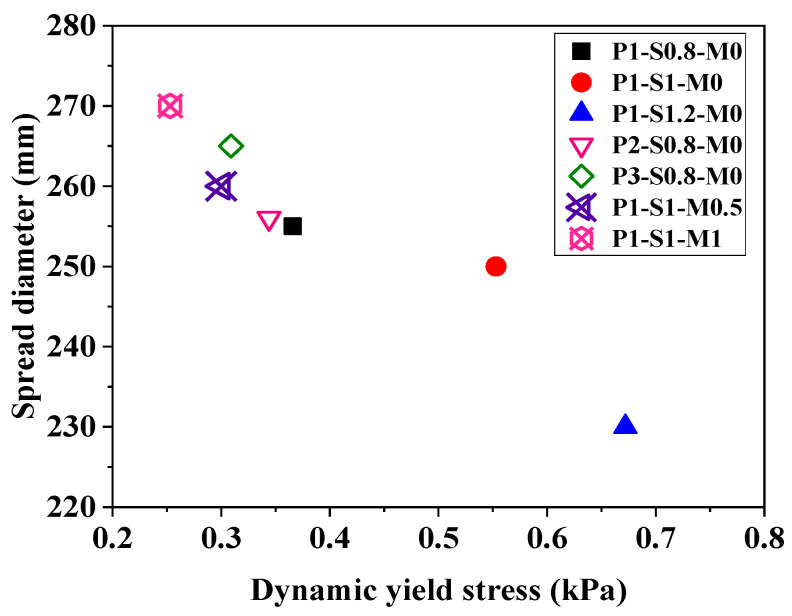
The relationship between the dynamic yield stress and the spread diameter of AAFS mortar.

**Figure 8 materials-15-01969-f008:**
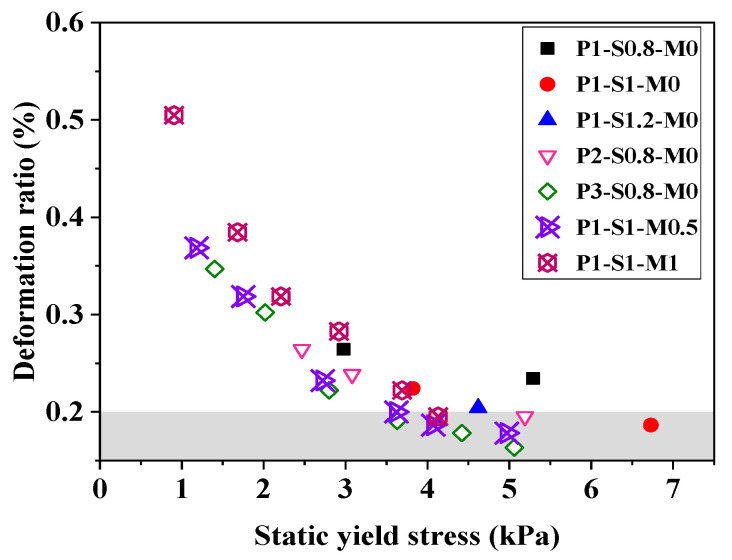
The relationship between static yield stress and deformation ratio of mortar.

**Figure 9 materials-15-01969-f009:**
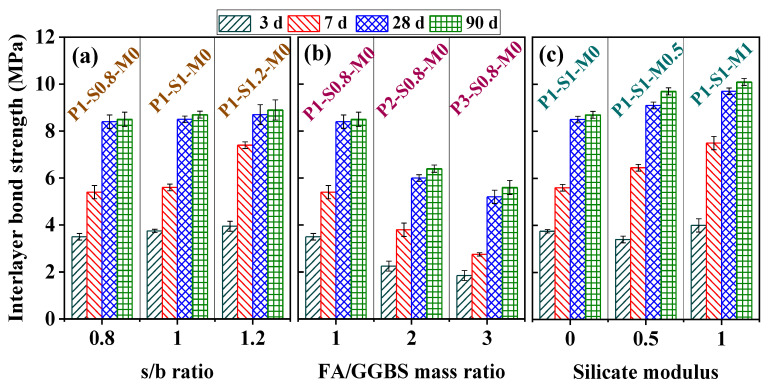
Interlayer bond strength of 3D-AAFS mortar. (**a**) effect of s/b ratio, (**b**) effect of FA/GGBS mass ratio, (**c**) effect of silicate modulus.

**Figure 10 materials-15-01969-f010:**
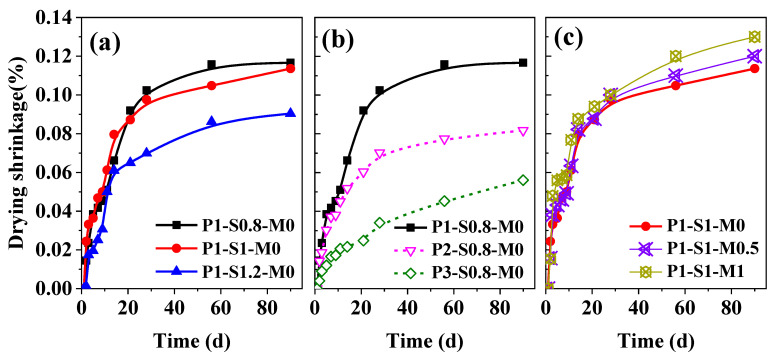
Drying shrinkage of samples with time. (**a**) effect of s/b ratio, (**b**) effect of FA/GGBS mass ratio, (**c**) effect of silicate modulus.

**Figure 11 materials-15-01969-f011:**
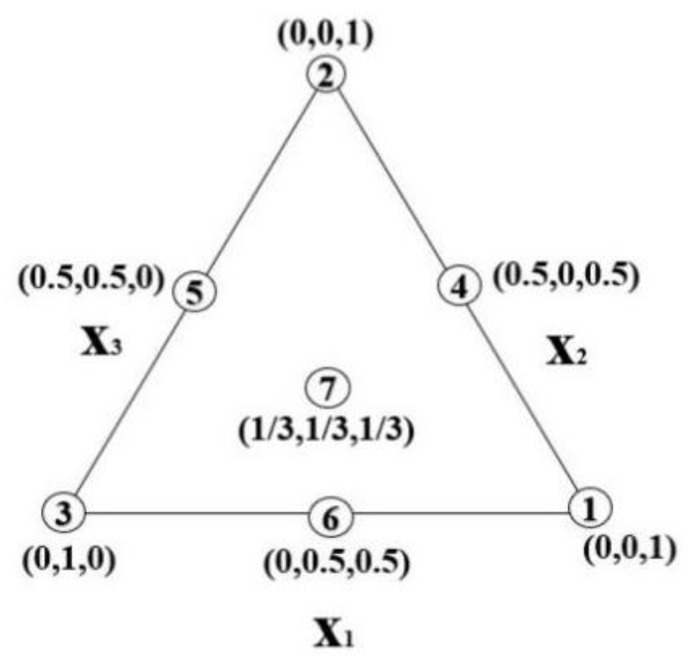
Seven test points of the simplex centroid design method.

**Figure 12 materials-15-01969-f012:**
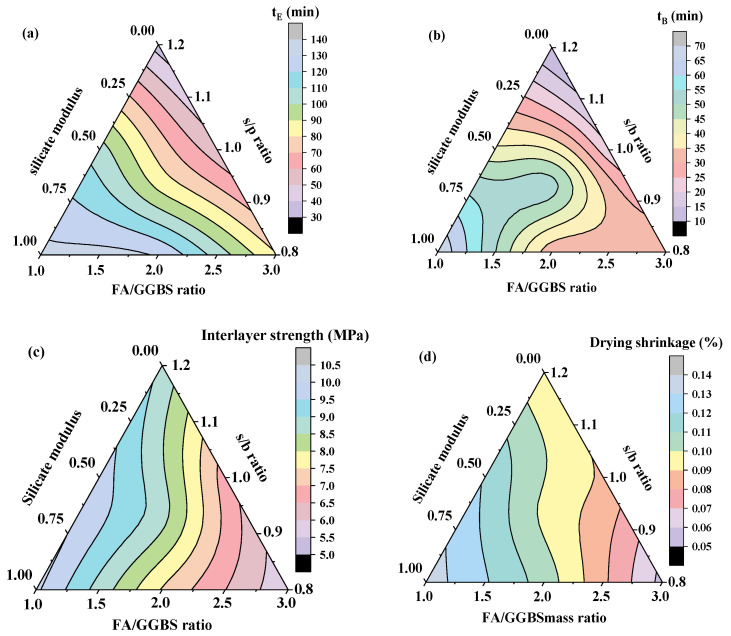
The contours of properties of 3D-AAFS mortar. (**a**) t_E_ contour, (**b**) t_B_ contour, (**c**) interlayer bond strength contour, and (**d**) drying shrinkage contour.

**Figure 13 materials-15-01969-f013:**
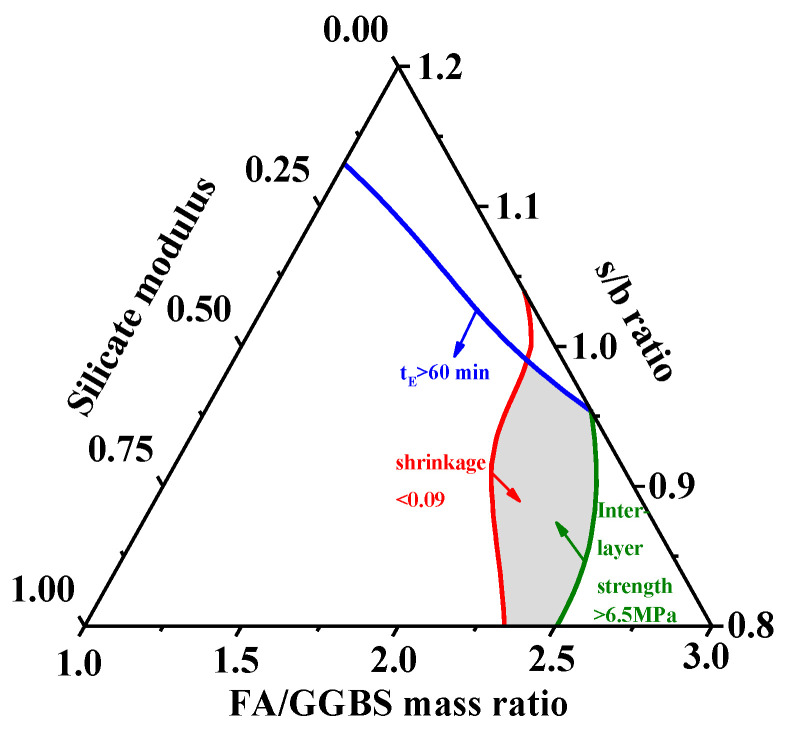
Optimization of the mix proportion of AAFS mortar for 3D printing manufacture.

**Table 1 materials-15-01969-t001:** Chemical compositions of GGBS and FA (wt.%).

	SiO_2_	CaO	Al_2_O_3_	Fe_2_O_3_	MgO	SO_3_	K_2_O	Na_2_O	LOI
GGBS	32.84	38.92	13.33	0.30	9.67	1.47	0.83	0.24	2.4
FA	59.38	2.16	29.74	3.80	1.36	0.83	0.32	0.42	1.99

**Table 2 materials-15-01969-t002:** Mix proportions of the AAFS mortars.

Sample	Mass Fraction (wt.%)			FA/GGBS Mass Ratio	s/b Ratio	Silicate Modulus	Alkali Dosage (Na_2_O wt.%)
	FA	GGBS	Sand				
P1-S0.8-M0	27.8	27.8	44.4	1	0.8	0	3.5
P1-S1-M0	25.0	25.0	50.0	1	1.0	0	3.5
P1-S1.2-M0	22.7	22.7	54.5	1	1.2	0	3.5
P2-S0.8-M0	37.0	18.5	44.4	2	0.8	0	3.5
P3-S0.8-M0	41.7	13.9	44.4	3	0.8	0	3.5
P1-S1-M0.5	25.0	25.0	50.0	1	1	0.5	3.5
P1-S1-M1	25.0	25.0	50.0	1	1	1	3.5
P1-S0.8-M1	27.8	27.8	44.4	1	0.8	1	3.5
P2-S0.8-M0.5	37.0	18.5	44.4	2	0.8	0.5	3.5
P2-S1-M0	33.3	16.7	50.0	2	1	0	3.5
P1.7-S1.1-M0.3	30.0	17.6	52.4	1.7	1.1	0.3	3.5

**Table 3 materials-15-01969-t003:** Static rheological parameters of samples obtained by the Roussel model.

Sample	t_s,0_ (kPa)	*A*_thix_ (kPa/min)	R^2^
P1-S0.8-M0	0.728	0.233	0.926
P1-S1-M0	1.021	0.291	0.987
P1-S1.2-M0	1.220	0.342	0.993
P2-S0.8-M0	0.704	0.147	0.914
P3-S0.8-M0	0.583	0.0742	0.991
P1-S1-M0.5	0.394	0.0739	0.979
P1-S1-M1	0.218	0.0675	0.969

**Table 4 materials-15-01969-t004:** The t_E_ and t_B_ of 3D-AAFS mortars.

	t_E_ (min)	t_B_ (min)
P1-S0.8-M0	60	30
P1-S1-M0	45	15
P1-S1.2-M0	35	10
P2-S0.8-M0	70	30
P3-S0.8-M0	80	35
P1-S1-M0.5	110	40
P1-S1-M1	115	60

**Table 5 materials-15-01969-t005:** The actual value and linear substitution of three factors.

Number in Figure 11	Sample	Actual Value of Factors			Linear Substitution of Factors		
		FA/GGBS Ratio (z_1_)	s/b Ratio (z_2_)	M_s_ (z_3_)	FA/GGBS Ratio (x_1_)	s/b Ratio (x_2_)	Ms (x_3_)
1	P1-S0.8-M1	1.0	0.8	1.0	0	0	1
2	P3-S0.8-M0	3.0	0.8	0.0	1	0	0
3	P1-S1.2-M0	1.0	1.2	0.0	0	1	0
4	P2-S0.8-M0.5	2.0	0.8	0.5	0.5	0	0.5
5	P2-S1-M0	2.0	1.0	0.0	0.5	0.5	0
6	P1-S1-M0.5	1.0	1.0	0.5	0	0.5	0.5
7	P1.7-S1.1-M0.3	1.7	1.1	0.3	1/3	1/3	1/3

**Table 6 materials-15-01969-t006:** The properties of selected samples according to the simplex centroid design method.

No.	Sample	t_E_ (min)	t_B_ (min)	Interlayer Bond Strength at 90 Days (MPa)	Drying Shrinkage at 90 Days (%)
1	P1-S0.8-M1	130	70	10.1 ± 0.07	0.135 ± 0.012
2	P3-S0.8-M0	80	35	5.6 ± 0.07	0.056 ± 0.008
3	P1-S1.2-M0	35	10	8.9 ± 0.07	0.0904 ± 0.005
4	P2-S0.8-M0.5	135	30	7.4 ± 0.07	0.107 ± 0.010
5	P2-S1-M0	55	20	6.7 ± 0.28	0.09 ± 0.006
6	P1-S1-M0.5	110	40	9.7 ± 0.14	0.12 ± 0.011
7	P1.7-S1.1-M0.3	80	55	8.8 ± 0.54	0.095 ± 0.008

## Data Availability

The data presented in this study are available on request from the corresponding author.

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
