# Peer review of "Factors Influencing the Properties of Extrusion-Based 3D-Printed Alkali-Activated Fly Ash-Slag Mortar"

_materials, 2022, doi:10.3390/ma15051969_

Round 1

Reviewer 1 Report

This paper is very well written and only minor revisions have to be made. I would have liked to see some discussions on the mechanisms behind the observations made. For example, why would a higher FA/GGBS ratio lead to longer tE & tB, etc. What are the physical and chemical mechanisms involved? At the moment, I see only descriptions of the trends observed without any in-depth explanations. Please see the annotated manuscript for other minor comments.

Author Response

Sincere thanks also go to the responsible and kind reviewer for helping us improve our manuscript in both scientific and linguistic aspects. 

Point 1

Line 93 steel fibre?

Response:

We have checked the sentence and confirmed that the work [Lim, J.H.; Panda, B.; Pham, Q.C. Constr. Build. Mater. 2018, 176, 690-699] indeed used steel cables to improve the flexural characteristics of 3D printed geopolymer composites.

Point 2

Line 122 Insert citation

Response:

We have added the citation in the revised manuscript.

Point 3

Line 143 Please also provide LOI

Response:

We have added the LOI data into Table1.

Point 4

Line 186 Would be useful to provide some details as to how the samples were cur, i.e. from which part of printed blocks, etc.

Response:

We have added the details of the sampling regime employed for the drying shrinkage test as follows:

Immediately after printing, both ends of the printed block were cut off, and the middle part of block (285×75×75 mm3) was used for the drying shrinkage test, and the sample was cured at 25±2 °C and 50±5% relative humidity. 

Point 5

Figure 2, please provide a scale bar

Response:

We have added the scale bars in Figure 2.

Point 6

Line 204 Was there a reference length comparator you used?

Response:

We didn’t use a reference length comparator in the drying shrinkage test. After being placed in the shrinkage frame, the 3D printed mortar had not moved during the whole measurement period (90 days). We measured the evolution of drying shrinkage strain of printed mortar in-situ by directly recording the readings of electronic dial gauge at designated curing times.

Point 7

Line 291 Why use 200 mm as a threshold?

Response:

As we mentioned in Section 2.3.1, a spread diameter less than 200 mm cannot be continuously extruded from the 3D printer we applied, therefore chose the 200 mm of spread diameter as a threshold to evaluate the extrudability. Actually, the threshold of 200 mm is close to the threshold of extrudability (192.5 mm) suggested by the previous work [Zhang Y., Zhang Y.S., She W., Yang L., Lin G.J., Yang Y.G.. Constr. Build. Mater. 2019, 201, 278-285.]

Point 8

Line 293, Again, why use 0.2%

Response:

In our previous work [Yuan, Q.; Li, Z.M.; Zhou, D.J.; Huang, T.J.; Huang, H.; Jiao, D.W.; Shi, C.J. Constr. Build. Mater. 2019, 227, 116600.], we detailed the reason for the choice in this threshold. According to a large number of experiments, we found that when the deformation of 20-layers load measured by this method is less than 0.2%, the printed mortar would easily collapse or the bottom layers would severely deform during the printing process. Therefore, we used 0.2% as a threshold to evaluate the buildability of the printed mortar in this study.

Point 9

Would be useful to plot interlayer bond strength vs buildability/printability to see the correlations in between.

Response:

Thanks for the review’s valuable suggestion. Actually, we have tried to correlate the buildability, printability, or rheological properties with the interlayer bond strength of 3D printed mortars. However, it was harder than expected for this study.

As we discussed in Section 3.2, the buildability/printability of 3D printed mortar greatly depends on its structural buildup rate. Many studies discussed the relationship between the structural buildup rate of fresh cement-based material and the interlayer bond strength of the hardened material. Figures R1(a)~(b) (please see in the attachment) plot the structural buildup rates of 3D printed mortars we investigated with their interlayer bond strength. It can be seen from Fig. R1(a) that there is not a simple relationship between Athix and interlayer bond strength due to the existence of three variables of mix proportion parameter. Fig. R1(b) shows a negative relationship between Athix and interlayer bond strength of mortars with different s/b ratios but the same FA/GGBS ratio and silicate modulus. However, this relationship is positive for mortars with different FA/GGBS ratios but the same s/b ratio and silicate modulus as shown in Fig. R1(c). Moreover, Fig. R1(d) provides the correlation for mortars with different silicate modulus but the same FA/GGBS ratio and s/b ratio, indicating that the relationship is negative.

Therefore, the relationships between Athix and interlayer bond strength are different in different situations. It was hard to illustrate this relationship based on the experimental data of this paper. This problem will need to be further gone into.

Point 10

I would have liked to see some discussions on the mechanisms behind these observations, i.e. why would a higher FA/GGBS ratio lead to longer tE & tB, etc. What are the physical and chemical mechanisms involved? At the moment, I see only descriptions of the trends observed without any in-depth explanations.

Response:

We have improved the sentences in this section and added more discussion about the physical and chemical mechanisms behind the experimental results. Please see the revised manuscript.

Reviewer 2 Report

There are grammar and typographic errors. Please correct these errors and further improve the language.

In this paper, how do the authors use 3D printed cementitious material? Author should present based on the standard.

The novelty of this work must be presented in last paragraph of Introduction and Conclusion sections very clear.

how do the authors use convolution method to analyze?

The results and figures are appropriate however; author should add more physical explanation for the observed results.

How to verify the accuracy and correctness of the analysis.

Conclusion section should modify and write as paragraph.

some results in that section should be merge.

Author Response

Authors would like to thank the reviewer for helping us improve the quality of our manuscript. The detailed responses to the review comments are listed below.

Point 1: There are grammar and typographic errors. Please correct these errors and further improve the language.

Response: We have made a complete language check throughout the paper and corrected the grammar and typographic errors.

Point 2: In this paper, how do the authors use 3D printed cementitious material? Author should present based on the standard.

Response: In this paper, we only investigated the fresh state and hardened state properties of 3D-printed mortar. However, the utilization of 3D-printed cementitious material was not involved in this study. According to the study results of this paper, we will try to use the 3D printed alkali-activated material in engineering practice in our further research.

Point 3: The novelty of this work must be presented in last paragraph of Introduction and Conclusion sections very clear.

Response: We have improved the sentences in the last paragraph of Introduction and Conclusion to highlight the novelty of this work as follows:

<In the last paragraph of Introduction>

For a full understanding of the relationship between the composition of 3D print-ed alkali-activated fly ash/slag (3D-AAFS) mortar and its properties, this study comprehensively investigated the influences of the sand-to-binder ratio, the relative proportion of FA-GGBS precursors, and the silicate modulus of the activator on the print-ability, interlayer bond strength and volume stability of 3D-AAFS mortar for the first time. Moreover, a simple centroid design method was developed for mix proportion-ing of extrusion-based 3D-AAFS mortar to strike a balance among printability, inter-layer bond strength, and volume stability. This study enriched the mix design concept for 3D printed alkali-activated materials. 

<In Conclusion>

A simple centroid design method was developed for mix proportioning of extrusion-based 3D printed AAFS mortar for the first time, which took printability, interlayer bond strength, and drying shrinkage into consideration at the same time. By restricting the fresh-state and hardened-state requirements, the optimum mix proportion of 3D-AAFS mortar can be obtained using this method. 

Point 4: how do the authors use convolution method to analyze?

Response: We did not use any convolution method in this study.

Point 5: The results and figures are appropriate however; author should add more physical explanation for the observed results.

Response: We have improved the sentences in the Results and discussion section, and added more discussion about the physical and chemical mechanisms behind the experimental results. 

Point 6: How to verify the accuracy and correctness of the analysis.

Response: In order to ensure the accuracy and correctness of the analysis, we repeated each test of property several times. For example, the rheological tests, flow table test, drying shrinkage test, and the interlayer bond strength test for each specimen were repeated three times and the experimental data presented in this paper was the average of results of the three times. Moreover, we also inserted the error bar in all of the figures of this paper to further clarify the uncertainty of measurement.

Point 7: Conclusion section should modify and write as paragraph. some results in that section should be merge.

Response: We have improved the sentences in the conclusion section. The three conclusions corresponded to the three main contents of this paper, i.e., the printability, the hardened-state properties, and the simple centroid design method. The paper focuses on three mixed proportion parameters and four properties, the readability will be reduced if the conclusions are written as a paragraph.

Reviewer 3 Report

It is highly recommended for publication in Materials.

1) All the resented results were well interpreted and concluding remarks were justified and supported by the results.

2) The quality of the presentation was very good.

3) The paper has great scientific soundness.

Author Response

Many thanks for your positive comments. It is my great honor to receive your recommendation.

Reviewer 4 Report

In this paper, authors present the results from an investigation regarding the effects of three mix proportion parameters of 3 D printed alkali-activated fly ash/slag (3D-AAFS) mortar. It is about the influences of the sand-to-binder ratio, the relative proportion of FA-GGBS (fly ash/ground granulated blast-furnace slag) precursors, and the silicate modulus of the activator on the printability, interlayer bond strength and volume stability of 3D-AAFS. The subject of the manuscript falls within the journal field. The manuscript is based on a solid documentation (75 references), it is well written, and it is interesting for the engineers and researchers in the field. The manuscript is articulated on two main parts (excluding the Introduction and Conclusions), witch describe firstly the experimental program, including the presentation of the raw materials and the tests series, and secondly, the presentation of the results and analysis which is supported and argued by an adequate theoretical and graphic material. Conclusions are clear and consistent with the analysis of the results, and they are useful for the field.     

Recommendation: accept following a careful English revision by a native speaking.

Author Response

Author would like to thank the reviewer for the positive comment. 

Point 1: accept following a careful English revision by a native speaking.

Response: We have made a complete language check throughout the paper and improved the language with the help of a native speaker.

Reviewer 5 Report

Authors reported the Factors influencing the properties of extrusion-based 3D 2 printed alkali-activated fly ash-slag mortar. This is an interesting work and it can accepted after minor revision.

  • Provide some numerical results in abstract
  • Rheological behaviour -Improve the discussion with reason and justification. Authors provided the observation in most of the places.
  • Why the increase in the s/b ratio only improves the interlayer bond strength 318 at the early stage, but it insignificantly influences the final strength?
  • Table 6- how many readings are taken? What is the error?
  • Check the language of the paper thoroughly
  • How the chemical composition was obtained in Table 1. Chemical compositions of GGBS and FA (wt.%).
  • Provide the ASTM standard followed for all the test in experimental details.

Author Response

Many thanks for the reviewer's comment. The detailed responses to the review comments are listed below.

Point 1: Provide some numerical results in abstract

Response: We have added some numerical results in the abstract as follows:

Although increasing the FA/GGBS mass ratio from 1 to 3 led to a reduction of 35% in the interlayer bond strength, it decreased the shrinkage strain by half. 

Point 2: Rheological behaviour -Improve the discussion with reason and justification. Authors provided the observation in most of the places.

Response: We have added more discussion about the physical and chemical mechanisms of rheological behavior.

Point 3: Why the increase in the s/b ratio only improves the interlayer bond strength at the early stage, but it insignificantly influences the final strength?

Response: Many factors can influence the interlayer bond strength of 3D printed mortar, such as roughness of substrate, cohesion and friction coefficients of the interface, moisture content and saturation level of the substrate, the porosity of the substrate [V.N. Nerella, S. Hempel, V. Mechtcherine, Construct. Build. Mater. 205 (2019) 586-601]. The increase of s/b may make different influences on the cohesion and friction coefficients of interface or porosity of the 3D printed mortar at the early stage and at the later stage, which improves the interlayer bond strength at the early stage, but it insignificantly influences the final strength. However, the underlying mechanism of interlayer bond strength of 3D printed material is difficult. It can not be well explained based on the experimental results of this study and this would have made the paper a bit too expansive. This problem will need to be further gone into. The reviewer pointed out a valuable suggestion and we will try to systematically study the mechanism using MIP, SEM, and NMR tests in our further study.

Point 4: Table 6- how many readings are taken? What is the error?

Response: In this paper, every measurement was repeated three times, and the results shown in Table 6 were the average of the readings obtained by the three repeated tests. We have added the error value into Table 6. 

Point 5: Check the language of the paper thoroughly

Response: We have made a complete language check throughout the paper 

Point 6: How the chemical composition was obtained in Table 1. Chemical compositions of GGBS and FA (wt.%).

Response: The chemical compositions were determined by X-ray fluorescence test, which was mentioned in Line 125.

Point 7: Provide the ASTM standard followed for all the tests in experimental details.

Response: We have added the ASTM standards followed in experimental and inserted the citations.
